# Genotoxicity Evaluation of Propyl-Propane-Thiosulfinate (PTS) from *Allium* genus Essential Oils by a Combination of Micronucleus and Comet Assays in Rats

**DOI:** 10.3390/foods10050989

**Published:** 2021-05-01

**Authors:** Antonio Cascajosa-Lira, María Puerto, Ana I. Prieto, Silvia Pichardo, Leticia Díez-Quijada Jiménez, Alberto Baños, Enrique Guillamón, Rosario Moyano, Verónica Molina-Hernández, Ángeles Jos, Ana M. Cameán

**Affiliations:** 1Área de Toxicología, Facultad de Farmacia, Universidad de Sevilla, Profesor García González n 2, 41012 Seville, Spain; aclira@us.es (A.C.-L.); mariapuerto@us.es (M.P.); anaprieto@us.es (A.I.P.); spichardo@us.es (S.P.); ldiezquijada@us.es (L.D.-Q.J.); angelesjos@us.es (Á.J.); 2DMC Research Center, Camino de Jayena, 82, 18620 Alhendín, Spain; abarjona@domca.com (A.B.); eguillamon@domca.com (E.G.); 3Department of Anatomy and Comparative Pathology and Toxicology, Faculty of Veterinary Medicine, University of Cordoba, Campus de Rabanales, 14014 Cordoba, Spain; r.moyano@uco.es (R.M.); b62mohev@uco.es (V.M.-H.)

**Keywords:** micronucleus, comet assay, genotoxicity, organosulfur compounds, propyl-propanethiosulfinate

## Abstract

Propyl-propanethiosulfinate (PTS) is a component of *Allium* essential oils. This organosulfur molecule can be used as a feed additive to decrease the appearance of bacterial resistances caused by the residues of antibiotics. In previous in vitro genotoxicity studies, contradictory results were reported for PTS. In this work, the in vivo genotoxicity of PTS in male and female rats was assessed for the first time, following OECD (Organisation for Economic Co-operation and Development) guidelines. After oral administration (doses: 5.5, 17.4, and 55.0 mg/kg PTS body weight), a combination of the micronucleus (MN) assay (OECD 474) in bone marrow and the standard and enzyme-modified comet assay (OECD 489) was performed. After necropsy, histopathological studies were also carried out. The results did not show the in vivo genotoxicity of PTS at any doses assayed, revealed by the absence of increased MN, and DNA strand breaks or oxidative DNA damage in the standard and enzyme-modified comet assays. The histopathological study revealed that only the highest dose tested (55.0 mg/kg) in the liver and all dose groups in the stomach presented minimal pathological lesions in the organs studied. Consequently, the present work confirms that PTS is not genotoxic at the doses assayed, and it is a promising natural alternative to synthetic preservatives and antibiotics in animal feed.

## 1. Introduction

*Allium* species and their essential oils have been extensively used for medical applications and as a flavoring agent. Vegetables of the genus *Allium* (e.g., garlic or onion) are well known for their antioxidant and anticancer properties in humans [1]. In addition, other properties such as antimicrobial, antispasmodic, antiasthmatic, anti-amnesic, anti-inflammatory, hepatoprotective, neuroprotective, hypotensive, hypoglycemic, immunomodulatory urease/xanthine oxidase inhibitory, and prebiotic properties have also been described [2,3,4,5,6].

Organosulfur compounds (OSCs) are secondary metabolites of plants from the *Allium* genus (e.g., thiosulfinates and ajoenes) with biological activity, and they have a characteristic aroma. These molecules are generated when tissues of these plants are damaged. If so, alliinase degrades the initial compounds into new ones exerting biological activity [7] (Figure 1). Some properties of these OSCs can be valuable for the agri-food industry, including their antioxidant, antibacterial, antifungal, and anti-yeast activities [8,9,10,11,12,13]. Considering the increasing interest of consumers in the use of natural active compounds as an alternative of chemical agents [14], products made from the essential oil of garlic and rich in OSC have been released into the market. One of the most promising products is ProAllium^®^ (DOMCA, Granada, Spain), which is composed of a mixture of several compounds, mainly propyl thiosulfinate oxide (PTSO) and, in a lower proportion, of propyl-propanethiosulfinate (PTS). ProAllium^®^ has been successfully incorporated into films to produce active packaging, able to expand the shelf life of ready-to-eat food and to control their spoilage [11,15,16]. In addition, several uses of PTS have been investigated and patented in relation to the treatment of farm animals. In this sense, it could be used against parasites in cattle, sheep, and poultry [17], for the reduction of apicomplexa in aquatic animals [18], and as an antimethanogenic additive in ruminant feed [19]. Therefore, the effectiveness of the PTS has been verified in many uses, and it could be a good natural alternative to synthetic agents.

However, its use for these applications requires the characterization of its toxicological profile considering consumer exposure to this compound. In this regard, the European Food Safety Authority (EFSA) panel on Food Contact Materials, Enzymes, Flavorings, and Processing Aids (CEF) has published the “Recent developments in the risk assessment of chemicals in food and their potentials impact on the safety assessment of substances used in food contact materials” [20]. Additionally, the panel on additives and products or substances used in Animal feed (FEEDAP) has published “Guidance on the assessment of safety of feed additives for the consumers” [21]. In these publications the EFSA explains that it is mandatory to study genotoxicity, even in the predictable case of low exposure.

Toxicological data about *Allium* extracts and essential oil are still scarce, especially in vivo. However, subchronic toxicity has been studied through a 90-day oral assay of Proallium [22], whose main component is PTSO. In this study, the extract demonstrated a complete absence of signs of toxicity. In addition, the pure PTSO has also been administered orally to rats for 90 days in a study published by Lira et al. [23].

Previous in vitro genotoxicity and mutagenicity studies for PTS carried out in our laboratory reported contradictory results [24]. Negative results were observed in the Ames test and mouse lymphoma (MLA) assay for PTS. In contrast, the micronucleus (L5178Y Tk ± cells) and comet assays (Caco-2 cells) revealed that PTS produced positive results at the highest concentrations assayed. These discrepancies in the results obtained raise the concern about PTS genotoxicity and make it necessary to clarify its genotoxic potential. For this purpose, EFSA suggests three different assays to assess the genotoxicity: the in vivo MN test, the in vivo comet assay, or the transgenic rodent gene mutation assay. Following the 3Rs principles (Replaces, Reduce, and Refine), Bowen et al. [25] suggested a combined multi-end point in vivo assay, and they evaluated the combination of a bone marrow MN test with the application of the comet assay, and actually, this in vivo combination assay in several organs has been proposed for international genotoxicity testing groups [26].

Considering all these facts, in order to clarify the contradictory results obtained in in vitro genotoxicity studies with PTS, the present study aimed to carry out the in vivo genotoxic assessment for PTS by using the combined MN-comet assay in rats for the first time. The tissues selected were bone marrow for the MN assay [27], and stomach, liver, and blood cells for the comet assay [28]. Moreover, the potential oxidative DNA damage was evaluated by the enzyme-modified comet assay with the bacterial repair enzymes formamidopyrimidine DNA glycosylase (FPG) and endonuclease III (EndoIII), which release damaged purines and pyrimidines, respectively [29]. Finally, a histopathological study was carried out on the stomach and liver to obtain further information for the genotoxic assessment and to provide useful information to perform the risk assessment required by EFSA for PTS before its approval as an additive.

## 2. Materials and Methods

### 2.1. Chemical and Reagents

PTS (Cod. 0200026, batch 31/19, 91% purity) was developed, chemically isolated, and provided by DMC RC (Granada, Spain). Sodium chloride (NaCl), potassium chloride (KCl), ethylenediaminetetraacetic acid (EDTA), sodium ethylenediaminetetraacetate (Na_4_EDTA), HEPES, Giemsa, and Trizma^®^ base were purchased from Sigma Aldrich (Madrid, Spain). Bovine serum was purchased from Gibco (Biomol, Seville, Spain). Endonuclease III and formamidopyrimidine DNA Glycosylase were purchased from Biolabs Inc. Sybr Gold was purchased from Invitrogen.

### 2.2. Animal Hosting and Nourishing Conditions

The Ethics Committee on Animal Experimentation of the University of Sevilla approved this in vivo experiment (09/03/2016/026). Animals were humanely cared for in compliance with the guidelines for the protection of animals used for scientific purposes (Directive 2010/63 UE).

Eight-week-old Wistar female and male rats, strain RjHan:WI (type outbred rats), were obtained from the Centre for Animal Production and Experimentation of the University of Sevilla (Spain). Upon arrival, animals were weighted and then accommodated into polycarbonate cages with stainless steel covers. Later, the animals were acclimatized to the environmental conditions with a 12 h dark/light cycle, temperature of 23 ± 1 °C, and relative humidity of 55 ± 10% for 1 week before the experiment. During this time, the animals were provided certified pellet food (Harlan, 2014; Harlam Laboratories, Barcelona, Spain) and tap water ad libitum, free from any type of chemical contamination.

### 2.3. Treatment Schedules and Dose Levels

The selection of PTS doses was calculated while taking into account a preceding acute toxicity study carried out by our research group, in which rats (8–10 weeks old) were administrated decreasing doses of PTS to establish the maximum tolerated dose (MTD). According to the OECD guidelines for testing chemicals [30], the initial dose tested was 175.0 mg of PTS/kg b.w., which induced the death of one of the three animals, so the next dose was established at 55.0 mg of PTS/kg b.w. In this case, three consecutive animals survived; therefore, the MTD of PTS was set at 55.0 mg of PTS/kg. At that dose, no remarkable injury was detected in the treated rats, and no evidence of study-limiting toxicity was observed. Then, considering the ICH S2 Guidelines and the OECD protocols 474 [27] and 489 [28], the limit dose or MTD of 55.0 mg/kg b.w. was chosen as the highest dose, and two additional properly spaced dose levels (divided by √10) were also calculated. Consequently, the doses were set at 55.0, 17.4, or 5.5 mg of PTS/kg b.w.

After acclimation time, 28 male and 28 female rats were randomly divided into control or treatment groups: the negative control (C−) (five male and five female rats), the solvent control (C sol) (five male and five female rats) treated with water with 0.3% Tween 80, and the positive control (C+) (three male and three female rats) treated with 200 mg/kg b.w. of ethylmethanesulfonate (EMS). The three PTS-treated groups (five male and five female rats per group) were exposed to 55.0, 17.4, and 5.5 mg of PTS/kg b.w. The number of animals was selected according to OECD 474 [27] and OECD 489 [28], which state five animals per sex and per group, and which also allow the use of at least three animals of both sexes treated with a positive control.

Doses were dissolved in water with Tween 80 (final concentration 0.3%) in a final volume of 1.5 mL. Tween 80 was previously used in genotoxicity in vivo studies by Mellado-Garcia et al. [31] with the aim of better conveying alliaceous compounds with an oily nature. All groups were dosed at 0, 24, and 45 h via gavage using a stomach tube (Vygon, Ecouen, France) and were sacrificed 3 h after the final dose administration for the combined comet assay and MN endpoints. The time points of PTS administrations, as well as the timing of sacrifice, were selected following the criteria of OECD guideline 489 and Bowen et al. [19].

After treatment, each rat was observed twice daily for clinical signs such as changes in the eye, fur, or skin, changes in gait and posture, and secretions.

### 2.4. Sample Collection

Samples of the liver and stomach were extracted, and a small part of each tissue was immediately handled, as described in Section 2.6, for the standard and enzyme-modified comet assay. Moreover, blood samples were extracted and preserved in Vacutainer^®^ sodium Heparin Tubes (Becton, Dickinson, Rutherford, NJ, USA). Furthermore, samples from the bone marrow of both femurs of each animal were gathered for the MN assay and were rapidly used to prepare smears (Figure 1). For the histopathologic study, sections of the liver and stomach were excised, rinsed with cold saline solution, and preserved in 10% buffered formalin for light microscopy, according to Llana-Ruiz-Cabello et al. [32].

### 2.5. Micronucleus Assay

The MN test was performed according to the OECD guideline 474 [27] and Diez-Quijada et al. [33]. The bone marrow of the femurs of the rats were mixed with a drop of fetal bovine serum. Therefore, slides were immersed in methanol, dried, and stained with Giemsa (Figure 1).

The polychromatic erythrocytes (PCE) among total erythrocytes (polychromatic and normochromic erythrocytes (NCE)) ratio and PCE for every NCE ratio were calculated by counting 500 erythrocytes per rat. The number of micronuclei found in immature erythrocytes was calculated by counting a total of 5000 micronucleated immature erythrocytes (MNPCE) per rat, and it was expressed as %MN.

### 2.6. Standard and Enzyme-Modified Comet Assay

Single cell suspensions from the liver and stomach were isolated following Diez Quijada et al. [33], and blood samples were isolated following Medrano-Padial et al. [25]. Both samples were placed on microscope slides. At first, slides were in lysis buffer (2.5 M of NaCl, 0.1 M of EDTA, 8 mM of Na_4_EDTA, and 9.9 µM of Trizma^®^ base) at 4 °C for at least 1 h. After this step, microscope slides intended for enzyme treatment were washed 3 times for 5 min with an enzyme buffer (buffer F) (pH 8; 0.2 mg/mL of bovine serum albumin; 0.5 mM of EDTA; 0.1 M of KCl; 40 mM of HEPES) Afterward, the agarose gels in slides were exposed to 30 µL of lysis buffer, buffer F, and buffer F comprising Endo III or FPG, and they were incubated in a metal box for 30 min at 37 °C. All microscope slides were subjected to electrophoresis for 20 min, at 0.81 V/cm up to 400 mA. The DNA released from the nucleus of the cells was immersed in PBS, water, 70% ethanol, and absolute before being stained with SYBR Gold (Figure 1).

At least 150 randomly selected nuclei per rat were analyzed with the image-analysis-software Comet Assay IV (Perceptive Instruments, UK). The %DNA in the tail, automatically acquired by the software, were used to define each of the comets analyzed. Endo III and Fpg sensitive sites were established by deducting the % of DNA in the tail after repair enzymes incubation. The medians of the scored comets were taken to describe each animal, and the means of each group of doses and controls are represented in Figure 2.

### 2.7. Histopathological Analysis

Liver and stomach samples were processed and observed under the microscope according to Diez-Quijada et al. [33]. A semiquantitative evaluation of the severity of lesions was scored as follows: 0, no significant lesions (0%); 1, minimal (<10%); 2, mild (11–25%); 3, moderate (26–50%); 4, severe (51–75%); and 5: very severe (>75%). Samples were individually examined by two unbiased and experienced observers, a veterinary pathologist, and an investigator (V.M-H. and R.M.).

### 2.8. Statistical Analysis

The results of the MN test are presented as the mean ± standard deviation (SD) for each group of animals, and a statistical analysis was performed using the analysis of variance (ANOVA) followed by Dunnett’s multiple comparison test. The results of the standard and enzyme-modified comet assays were calculated for each group as the mean ±SD of the medians. The distribution of the results was checked for normality utilizing the Kolmogorov–Smirnov test, and the total scores of the different groups were compared using the nonparametric Kruskal–Wallis test followed by Dunn’s multiple comparison test. For the histopathological study, the Kolmogorov–Smirnov test was used to assess if distributions were parametric. Groups were compared by applying the nonparametric Kruskal–Wallis test followed by Dunn’s multiple comparison test. *p* values < 0.05 were considered statistically significant. Analyses were carried out using Graph-Pad Prism software (Graph-Pad software Inc., La Jolla, San Diego, CA, USA).

## 3. Results

No clinical signs were observed during the entire course of the study after treating the rats with all levels of doses of PTS, water, or water with added Tween 80. However, rats treated with EMS presented piloerection and an attenuation in their normal physical activity.

The results obtained in the positive and negative control groups in others and recent studies [29] were similar to those obtained in the present work for comet and micronucleus assays.

### 3.1. Micronucleus Assay

In Table 1, the results of the in vivo MN assay of rats exposed to PTS are shown. When the results were compared to their respective controls, significant differences in PCE/NCE ratio were found, showing a decrease at the highest dose assayed (55 mg/kg) in males and females (*p* < 0.05). Considering the PCE/Total ratio, a significant reduction was also observed in males and females at 55 mg/kg (*p* < 0.05), and in females at 17.4 mg/kg (*p* < 0.05). Moreover, the % of MN in immature erythrocytes was not significantly changed in any groups of either sex. Rats exposed to 200 mg/kg of EMS showed remarkable reductions in PCE/NCE and PCE/total ratios, and a significant increase in the frequency of MN (positive control).

### 3.2. Standard and Enzyme-Modified Comet Assay

The %DNA in the tail results found in the standard and enzyme-modified comet assays are represented in Figure 3. No observed DNA stand breaks were recorded in the liver, stomach, and blood in rats treated with PTS (Figure 3A) at any doses assayed in either sex. Likewise, no remarkable changes were found in the frequency of Endo III or FPG-sensitive sites in any tissue assayed (Figure 3B,C). In contrast, significant elevations were clearly observed in rats treated with 200 mg/kg of EMS (positive control). The distribution of comet results with each treatment by presenting data for individual comet data is available as Appendix A.

### 3.3. Histopathological Study

The histopathological study of the stomach and liver did not show differences between males and females (data not shown). The scores obtained after the pathological evaluation of both the stomach and liver are shown in Figure 4.

Under light microscopy, the negative and solvent controls stomach samples showed no pathological signs, displaying a minimal flaking epithelium of the gastric mucosa (Figure 5A,B). The positive controls displayed lesions scored as mild, showing an evident flaking epithelium with focal necrosis of the apical gastric glands (Figure 5C). This group showed significant differences (*p* < 0.05) in the damage recorded in comparison to the negative and solvent control group (Figure 4). Occasionally, there were minimal inflammatory infiltrates with neutrophils and macrophages within the connective tissue of the submucosa layer. The PTS 55.0 mg/kg dose group presented minimal to mild pathological lesions in the glandular stomach, such as hypertrophy of the mucus cells and flaking epithelium (Figure 5), these changes being statistically significant (*p* < 0.05) compared to the negative and solvent controls (Figure 4). However, the PTS 17.4 and 5.5 mg/kg doses presented statistically significant minimal histopathological changes (Figure 4E,F). Occasionally, in the rats exposed to PTS 17.4 mg/kg, minimal hyperemia of the apical mucosa glands was observed (Figure 5E). Moreover, in the positive control and PTS 55.0 mg/kg groups, lesions were found in the nonglandular stomach characterized by vacuolation of the squamous epithelium in the mucosa associated with a mild to moderate pyogranulomatous inflammatory infiltrate and by a moderate pyogranulomatous inflammatory infiltrate in the submucosa (Figure 5H,I).

Livers from the negative and solvent controls showed no pathology findings apart from a normal physiological deposition of glycogen (Figure 6A,B). The lesions found in the positive controls were a mild multifocal distribution of hepatocyte degenerated in the surroundings of the central vein with occasional necrosis of hepatocytes (Figure 6C). Occasionally, there were areas filled with plasma and congestion. These lesions were significantly different (*p* < 0.05) in comparison to negative and solvent controls (Figure 4). Moreover, PTS doses of 55.0 mg/kg displayed minimal inflammatory infiltrates apart from a minimal centrilobular degeneration of hepatocytes (Figure 6D) showing statistically significant differences (*p* < 0.05) compared to the negative and solvent controls (Figure 4). However, PTS 17.4 and 5.5 mg/kg doses displayed minimal changes similar to the negative and solvent controls (Figure 6E,F).

## 4. Discussion

The interest in the genotoxicity evaluation of PTS is growing because of new potential applications in the food industry (feed additive, active in food packaging, etc.). Genotoxicity studies are one of the basic sets of toxicological studies included in the guidelines on the safety of feed additives, or substances employed in food contact materials, required by EFSA [22,23,24,25,26,27,28,29,30,31,32,33,34] before their approval.

As it has been mentioned before, previous PTS in vitro genotoxicity studies provided divergent results about its genotoxic potential [24]. Thus, PTS was not mutagenic in the Ames test (8.75–280 µM), in the presence or absence of S9, or in an MLA assay on L5178YTk+/− cells (0.9–15.65 µM) after 24 and 48 h of treatment. In contrast, positive results were found in the MN test in vitro on L5178YTk+/− cells at the highest concentration assayed (17.25 µM) without S9, as well as on its metabolites (+S9) (from 20 µM). Moreover, positive results were found in the standard comet assay in Caco-2 cells after 24 and 48 h, at the highest concentration assayed (280 µM). However, after the incubation with enzymes (Endo III or FGP), no increase in DNA damage was detected in the cells treated with PTS (24–48 h). In cases of contradictory results, such as the case of PTS, new in vivo genotoxicity studies of this compound are needed according to the recommendations of EFSA [34].

In this sense, the in vivo results reported in the present work confirm that PTS is not genotoxic in male and female rats orally exposed (55.0, 17.4, or 5.5 mg PTS/kg b.w.), according to the negative results observed for the MN and comet assays. Furthermore, no DNA oxidative damage was induced by PTS in any of the tissues investigated (blood, stomach, and liver) when the enzyme-modified comet assays were applied. In case of negative results in the in vivo genotoxicity studies, EFSA [34] indicates the importance of obtaining evidence of target cell exposure, and of considering other relevant tissues. In the present work, the significant reductions in the PCE/NCE ratios obtained in the rats treated with PTS confirmed the bone marrow exposure [27].

Regarding the experimental procedure performed in the present work, the combination of the in vivo MN and comet assays into a single rodent study is recommended by EFSA [23] and OECD protocols [28] in order to reduce the number of animals employed following the 3Rs principles (replace, reduce, and refine). Moreover, the sensitivity and specificity of the genotoxic damage increase, and the number of false-negative results also decrease [35]. Recent studies have revealed that oxidative stress on DNA bases is involved in many of the mechanisms of action of genotoxic substances [36]. For that reason, the use of DNA repair enzymes, especially Endo III and FPG, has been widely employed. This technique also improves the sensitivity of the in vivo comet assay by making it possible to identify potential oxidative DNA damage before the DNA break takes place, overcoming the potential false-positive results obtained in the standard assay [37]. Enzymes are also helpful to explore the further mechanism of action such as alkylation, or SBs, and they can provide information about the cellular response and how fast damage is repaired [38]. EFSA [34] emphasizes a careful evaluation of the results while taking into account all the information of the compound and its mechanism of action, and therefore, the enzyme-modified comet assay can provide us with a way in which the xenobiotic acts on the bases of DNA.

Histopathological examination of the most related tissues, the liver and stomach, was carried out in order to provide additional relevant evidence for the global toxicological profile and support to identify the possible target organs for PTS. Generally, rats treated with PTS (55.0, 17.4, and 5.5 mg/kg) showed very mild histopathological changes, and, generally, only minimal inflammatory processes in the liver and stomach were detected at the highest dose (55 mg/kg b.w.) in the liver and in all dose groups in the stomach. These results agree with the histopathological damage observed in the in vivo genotoxicity study performed with PTSO [31], in which there was no relevant damage in the same tissues, except for an increase in glycogen accumulation in the liver and a slight degeneration in the stomach at the highest dose tested (55.0 mg/kg). Regarding the selection of these organs for genotoxicity assessment and histopathological analysis, the liver is a primary site of xenobiotic metabolism and is often highly exposed to both parent substances and metabolites, and the stomach is a tissue of direct contact for oral administration [26,35].

To better understand the safety of PTS in vivo, the in vitro genotoxicity studies and our in vivo results need to be compared. Regarding the concentrations assayed previously by Mellado-García et al. [24], PTS had positive responses in vitro at 17.5 µM in the MN test and at 280.0 µM in the comet assay, in contrast to the negative results found in our study, even at the maximum concentration assayed of 55.0 mg/kg. This finding could be due to the fact that the in vitro tests were carried out under cytotoxic conditions. Similarly, propyl propane thiosulphonate (PTSO), the major component of ProAllium^®^, showed contradictory responses after the application of a battery of four in vitro genotoxicity assays [37], and later, the in vivo combined MN-comet assay in male and female rats demonstrated a lack of PTSO genotoxicity [31].

These contradictory results between the genotoxicity test in mammalian cell lines in vitro showing positive results, when compared to the negative findings in the in vivo assays of both OSC compounds, PTS and PTSO, could be due to several reasons [38,39]. The first if the different sensibility of the experimental model; hence, mammalian cells lines are highly sensitive to giving positive results, due to their scarcity in the metabolism, p53 function, and DNA repair capability. In fact, EFSA [39] indicated that over 90% of the in vitro positives were “unrelated positives” identified by these cells under in vitro conditions. Secondly, the concentrations tested in vitro are usually higher than the cytotoxicity threshold, a condition that is now required in mammalian cells, which contributes to the high frequency of false-positive results [38]. In addition, there are differences in the metabolism and/or bioavailability of the compound assayed in the target organ, because higher concentrations have been assayed in the in vivo assays. Regarding the first factor, when test substances were less toxic in vivo, detoxification processes are usually implicated [31,32]. In general, the detoxification system can play a fundamental role in the impact of genotoxic or mutagenic compounds in different ways: (a) by avoiding the formation of the final mutagen by metabolic sequestration of the promutagen or (b) by inactivation of the active metabolite after its creation [31]. In this sense, several authors have reported the metabolization of OSC present in the oil section of *Allium* sp. such as DADS (diallyl disulfide) by oxidation (cytochrome P450) to produce diallyl sulfoxide (DASO) and allyl sulfone (DASO2). Moreover, various in vivo studies have found that DAS (diallyl sulfide) and DADS can modulate the phase I and phase II metabolizing enzymes [40,41]. According to our hypothesis of the metabolization of OSC compounds as a possible detoxification route, Mellado-García et al. [31] reported the presence of degradation products of PTSO in the stomach of rats exposed, after the application of pyrolysis techniques, whereas in the liver, only one was identified in comparison to control rats, indicating the influence of the distribution and metabolism processes.

Further research focused on metabolism processes and toxicokinetic aspects of PTS are necessary to define metabolic derivates of this organosulfur compound and to assess its safety to be applied as a feed additive. In addition, following the guidance of EFSA for feed additives, the subchronic toxicity test should be performed through a 90-day assay.

## 5. Conclusions

The combined in vivo MN-comet assay protocol showed that PTS did not produce genotoxic effects in the stomach, liver, blood cells, and bone marrow of male and female rats orally (gavage) exposed to 5.5, 17.4, and 55.0 mg/kg b.w. In addition, no oxidative DNA damage was induced by PTS when the enzymes FPG and Endo III were applied in the nucleus isolated from the cells of the liver, stomach, and blood. The histopathological study revealed only minimal hypertrophy of the mucus cells and flaking epithelium in the stomach and slight lesions in the liver, and occasionally, minimal inflammatory infiltrates in rats exposed to the highest dose (55.0 mg of PTS/kg). Consequently, the absence of in vivo genotoxic effects of PTS supports its approval as a feed additive or active agent in food contact materials, although more studies are needed to obtain a global picture of its safety profile.

## Figures and Tables

**Figure 1 foods-10-00989-f001:**
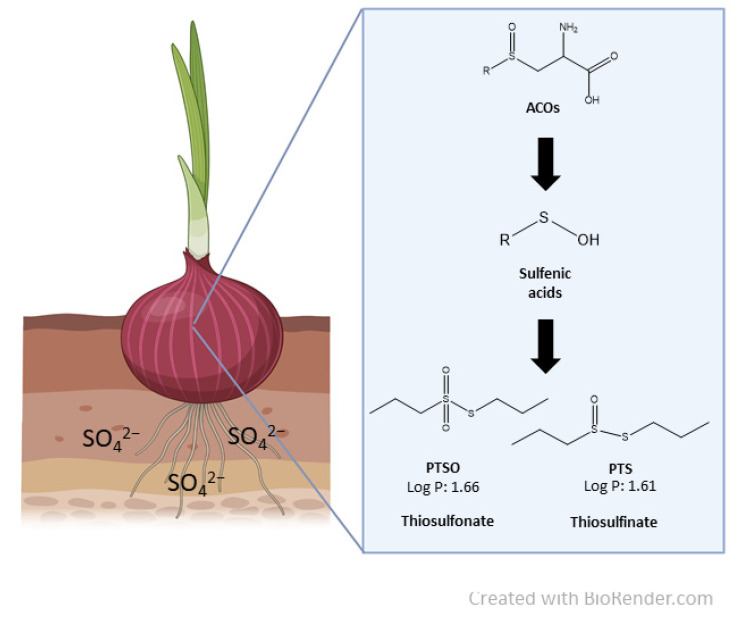
Scheme of the chemical structures and biochemical pathways of organosulfur compounds. ACOs: Alk(en)yl cysteine sulfoxides.

**Figure 2 foods-10-00989-f002:**
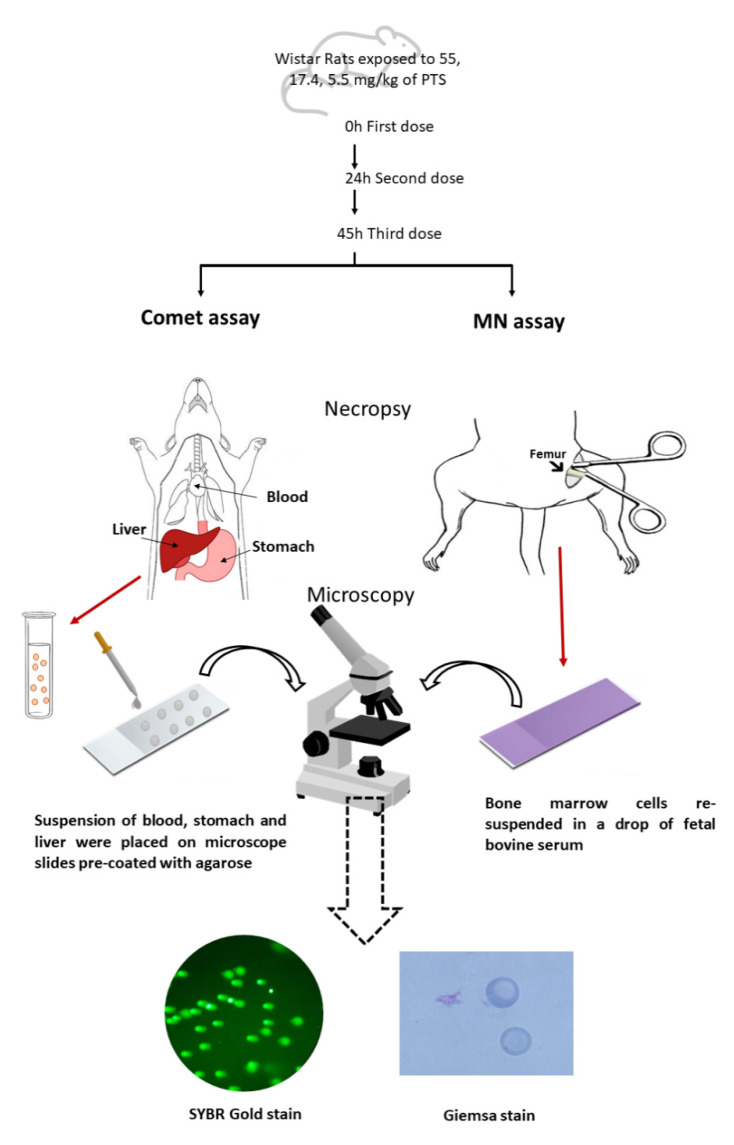
Micronucleus and comet assay schematic diagram on the same animal and its visualization under the optical microscope (fluorescence and bright field).

**Figure 3 foods-10-00989-f003:**
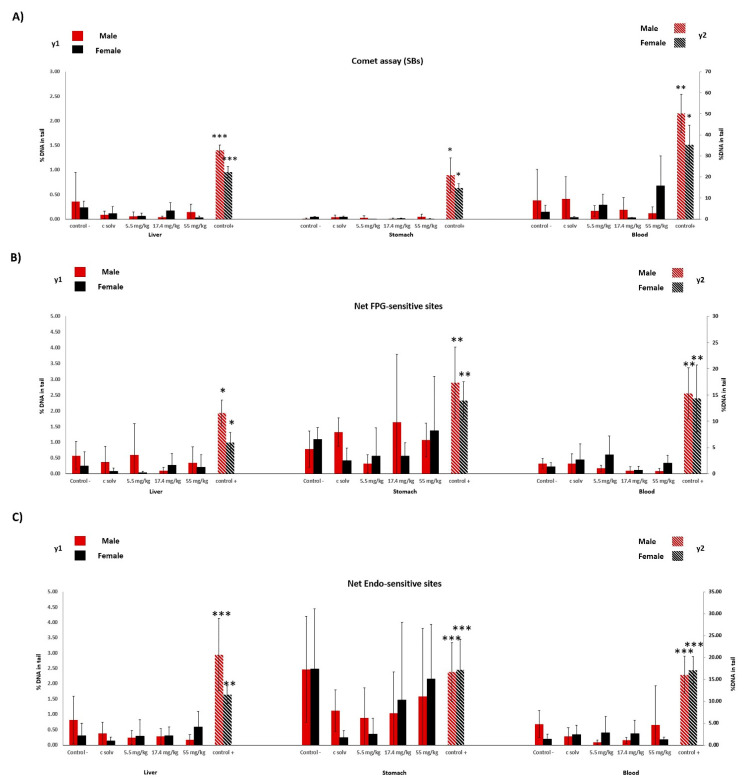
The % of DNA in the tail is representative of damage produced in the liver, stomach, and blood of male and female rats exposed to PTS and EMS (positive control). (**A**) Damage by the standard comet assay as strand breaks (SBs), (**B**) oxidative damage detected by the FPG enzymes, and (**C**) oxidative damage detected by Endo III enzyme. All values are expressed as the mean ± SD. The statistical test used was ANOVA or Kruskal–Wallis based on the normality test. Significant differences from the negative control were: * *p* < 0.05; ** *p* < 0.01; *** *p* < 0.001.

**Figure 4 foods-10-00989-f004:**
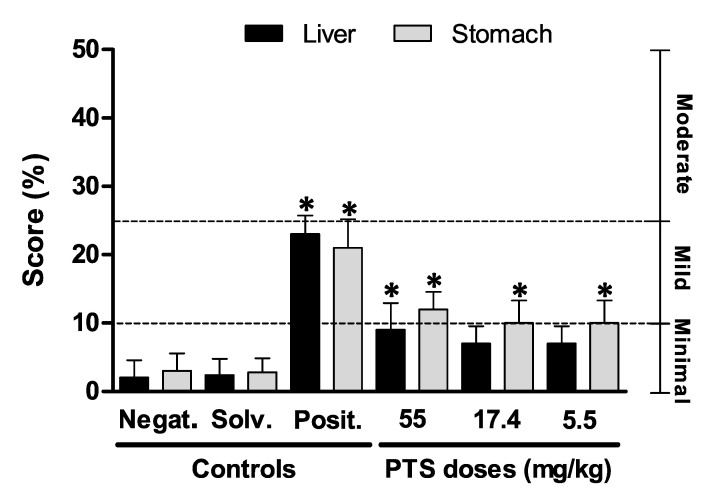
Mean ± standard deviation (SD) of the histopathological score recorded in the liver and stomach of female and male rats treated with PTS. * Statistical difference (*p* < 0.05) compared to the negative and solvent controls using the nonparametric Kruskal–Wallis test followed by Dunn’s multiple comparison test.

**Figure 5 foods-10-00989-f005:**
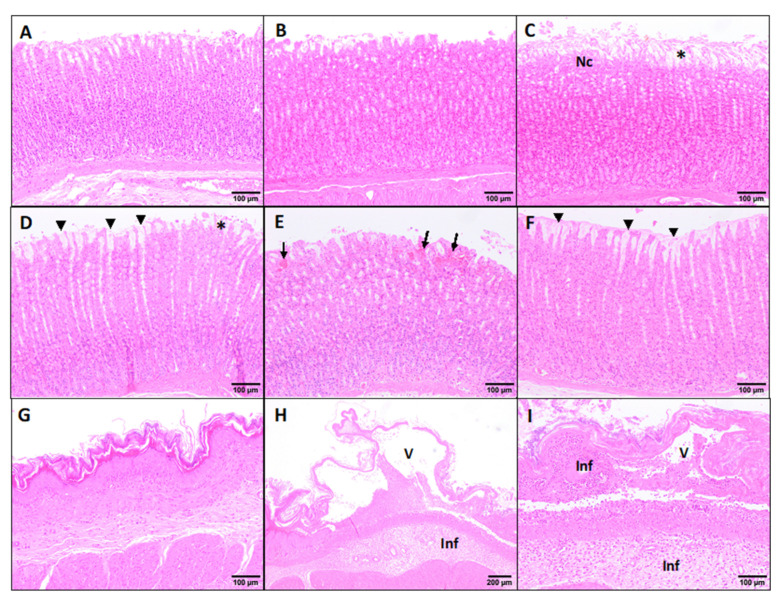
Histopathological changes in the stomach of rats exposed to PTS. (**A**–**C**) Glandular stomach of controls (negative, solvent, and positive, respectively); (**D**–**F**) Glandular stomach of PTS doses (55.0, 17.4, and 5.5 mg/kg, respectively); (**G**–**I**) Nonglandular stomach of negative control, positive control, and PTS 55.0 mg/kg dose, respectively. Hematoxylin and eosin stain. *: flaking epithelium of the gastric mucosa; Nc: focal necrosis of the apical gastric glands; arrow-head: hypertrophy of the mucus cells; arrow: occasional hyperemia of the apical mucosa glands; V: vacuolation of the squamous epithelium in the mucosa; Inf: inflammatory infiltrates in the mucosa and submucosa.

**Figure 6 foods-10-00989-f006:**
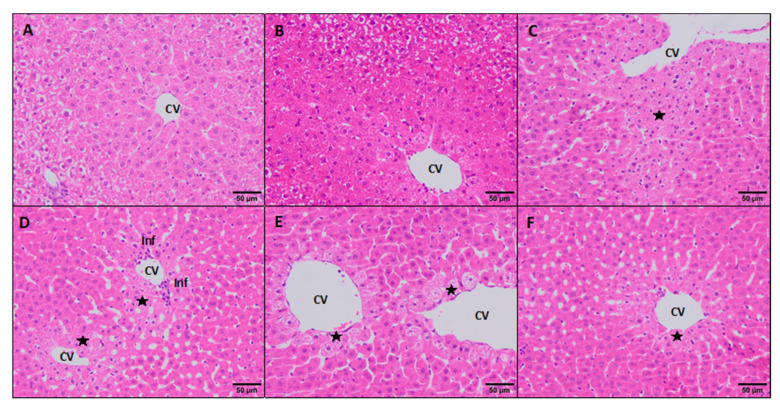
Histopathological changes in the liver of rats exposed to PTS. (**A**–**C**) Controls (negative, solvent, and positive, respectively); (**D**–**F**) PTS doses (55.0, 17.4, and 5.5 mg/kg, respectively). Hematoxylin and eosin stain. CV: central vein; ⋆: degenerated hepatocytes; Inf: inflammatory infiltrates.

**Table 1 foods-10-00989-t001:** Results of the micronucleus assay in rats exposed to PTS. Bone marrow cytotoxicity expressed as polychromatic erythrocytes (PCE) among total erythrocytes (normochromic erythrocytes (NCE) + PCE), ratio PCE among NCE, and the micronuclei induction expressed as % micronucleus (MN). Statistical analysis was performed using the analysis of variance (ANOVA) followed by Dunnett’s multiple comparison test. The values are expressed as the mean ± SD. Significant difference from the negative and solvent control (* *p* < 0.05; ** *p* < 0.01; *** *p* < 0.001).

Groups	Sex	N	Doses	PCE/NCE	PCE/Total	%MN
Negative Control	♂	5	-	1.72 ± 0.85	0.57 ± 0.04	0.57 ± 0.23
	♀	5	-	1.27 ± 0.07	0.61 ± 0.05	0.86 ± 0.16
Solvent Control	♂	5	-	1.74 ± 0.63	0.60 ± 0.09	0.49 ± 0.37
	♀	5	-	1.16 ± 0.34	0.53 ± 0.07	0.75 ± 0.24
Positive Control (EMS ^1^)	♂	3	200.0 mg/kg	0.54 ± 0.12 ***	0.35 ± 0.05 **	1.46 ± 0.72 *
	♀	3		0.53 ± 0.29 **	0.33 ± 0.12 **	2.53 ± 0.38 *
PTS	♂	5	5.5 mg/kg	1.31 ± 0.30	0.56 ± 0.05	0.46 ± 0.21
	♀	5		1.03 ± 0.43	0.49 ± 0.10	0.86 ± 0.09
	♂	5	17.4 mg/kg	1.14 ± 0.34	0.52 ± 0.08	0.72 ± 0.30
	♀	5		0.68 ± 0.36	0.38 ± 0.13 *	0.74 ± 0.06
	♂	5	55.0 mg/kg	0.89 ± 0.21 *	0.46 ± 0.06 *	0.46 ± 0.15
	♀	5		0.54 ± 0.28 *	0.33 ± 0.11 **	1.03 ± 0.21

^1^ EMS = ethylmethanesulfonate; ♂ = Males; ♀ = Females.

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
