# Peer review of "Genotoxicity Evaluation of Propyl-Propane-Thiosulfinate (PTS) from Allium genus Essential Oils by a Combination of Micronucleus and Comet Assays in Rats"

_foods, 2021, doi:10.3390/foods10050989_

Round 1
Reviewer 1 Report
The manuscript entitled “Genotoxicity evaluation of Propyl-propane-thiosulfinate (PTS) 2 by a combinate micronucleus and comet assays in rats” by Cascajosa-Lira et al aims at the investigation of the in vivo genotoxicity of PTS. This study is a follow-up study from the same group after finding inconclusive results following an investigation of the in vitro toxicity of this molecule. The present work comprises a substantial amount of work, using the comet assay (classical as well as the FPG- and EndoIII-modified assays), as well as the in vivo micronucleus assay. The authors also performed histopathological analyses on relevant organs of exposure.
Comments
- Minor grammatical and spelling errors are found throughout the manuscript. A careful proofreading of the manuscript, and some minor English revisions are necessary.
- The introduction would benefit from a description of the oral toxicity of allium genus extracts.
- The materials and methods section is severely lacking in necessary details. More information is needed in the chemicals and reagents section on which products were provided by each supplier.
- The selection of doses was appropriate. In the materials and methods section, it is stated on line 116 that the initial dose resulted in the death of one animal. It should be specified how many animals were treated at this dose.
- The description of the comet assays (classical and enzyme modified assays), as well as the MN test is not sufficient. Considerable more detail needs to be added to these sections. What were the migration conditions and voltage gradients? How many comets were scored? What software was used for the scoring? How were net-FPG and net-EndoIII sites calculated? How many cells were counted for the micronucleus assay?
- Are historical data for negative controls available for the micronucleus test? The solvent control for female rats seems high with some overlap with the positive control. How many cells were counted?
- The results from the comet assay are surprising. It is odd that the positive control did not generate more tail DNA considering the short time following the last exposure. As well, the percentage of tail DNA in the treated organs is very low, especially for the stomach. I am wondering if the migration conditions were not optimal. The migration time and voltage gradient needs to be specified. Are historical data for the negative controls available?
- The graphs for the comet assay are very difficult to interpret. An interrupted or broken y-axis is needed since the difference between the positive control and samples is so large. This would help to see any variation or trends in tail DNA.
- It would also be interesting to see the variation in the distribution of comets with each treatment by presenting data for individual comets.
- It is not clear how the results for the comet assay were calculated. Were medians for each slide and animal calculated? Are the data presented the means of the medians for each group?
- In light of the high level of intestinal exposure, it would have been interesting to investigate intestinal effects as well, including comet and histopathological assays.
- It is interesting that the histological examination showed slight increases in damage in the liver and stomach at all concentrations tested (although only the highest concentration was statistically significant). It would have been interesting to quantify serum markers of liver damage (ALT for example)
- In the authors’ previous in vitro study, concentrations of PTS giving positive results in the comet assay gave rise to significant toxicity (~IC50). Could this be a false positive result based on high levels of cytotoxicity at this concentration?
- Line 336: The authors indicate that 55.0 mg/kg corresponds to 331.0 μM. Does this represent serum concentration? How was this concentration calculated? Without appropriate quantification, it is extremely difficult to extrapolate. How much of the dose was absorbed? Does the concentration represent organ exposure? I would suggest to remove this extrapolation.
- Simply as a point of interest, what could be the concentration of PTS in feed or other commercial purposes necessary for antimicrobial activity compared to exposure concentrations? How does this amount correlate with the concentrations tested in this study?
Overall, the present manuscript represents a substantial amount of work. The study was well-designed with an appropriate number of animals, appropriate concentrations and an appropriate treatment schedule. I agree with the authors that the in vivo results reported in the present work confirm that PTS is not genotoxic in vivo in the rat model.
Author Response
Reviewer 1
The manuscript entitled “Genotoxicity evaluation of Propyl-propane-thiosulfinate (PTS) 2 by a combinate micronucleus and comet assays in rats” by Cascajosa-Lira et al aims at the investigation of the in vivo genotoxicity of PTS. This study is a follow-up study from the same group after finding inconclusive results following an investigation of the in vitro toxicity of this molecule. The present work comprises a substantial amount of work, using the comet assay (classical as well as the FPG- and EndoIII-modified assays), as well as the in vivo micronucleus assay. The authors also performed histopathological analyses on relevant organs of exposure.
RESPONSE: First of all, thank you for your thorough revision. All your suggestions have been taken into account and the manuscript has been improved.
Comments
- Minor grammatical and spelling errors are found throughout the manuscript. A careful proofreading of the manuscript, and some minor English revisions are necessary.
RESPONSE: Thank you for your comment. English has been revised throughout the text.
- The introduction would benefit from a description of the oral toxicity of allium genus extracts.
RESPONSE: Thank you for your suggestion. A description of the oral toxicity of allium genus extract has been implemented in the introduction: “Toxicological data about Allium extracts and essential oil are still scarce, especially in vivo. However, subchronic toxicity has been studied through a 90-day oral assay of Proallium, which main component is PTSO [17]. In this study, the extract demonstrated a complete absence of signs of toxicity. In addition, the pure PTSO has also been administered orally to rats for 90 days in the study published by Cascajosa Lira et al. [18].”
Consequently, two new references have been added, nº 17 and 18, and the references section has been modified.
- The materials and methods section is severely lacking in necessary details. More information is needed in the chemicals and reagents section on which products were provided by each supplier.
RESPONSE: Thank you for your comment. You are right. More information has been added in the chemicals and reagents section. The new paragraph is:
“PTS (Cod. 0200026, batch 31/19, 91% purity) was developed, chemically isolated, and provided by DMC RC (Granada, Spain). sodium chloride (NaCl), potassium chloride (KCl), Ethylenediaminetetraacetic acid (EDTA), Sodium ethylenediaminetetraacetate (Na4EDTA), HEPES, Giemza, and Trizma ® base were purchased from Sigma Aldrich (Madrid, Spain). Bovine serum were purchased form Gibco (Biomol, Seville, Spain). Endonuclease III and, Formamidopyrimidine DNA Glycosylase were purchased from Biolabs Inc. Sybr Gold were purchased from Invitrogen”.
- The selection of doses was appropriate. In the materials and methods section, it is stated on line 116 that the initial dose resulted in the death of one animal. It should be specified how many animals were treated at this dose.
REPONSE: Thank you for your suggestion. The number of total animals administrated with 175 mg/kg were 3. This data has been implemented in the text (line 124). The selection of the dose was carried out according to the guideline OECD 425. This guideline provides the DL50, so it has been adapted in order to obtain the MTD. Briefly, this is a sequential study giving the dose to 1 animal in each step. If the animal dies the dose should be reduced and if the animal survives the dose should be increased.
- The description of the comet assays (classical and enzyme modified assays), as well as the MN test is not sufficient. Considerable more detail needs to be added to these sections. What were the migration conditions and voltage gradients? How many comets were scored? What software was used for the scoring? How were net-FPG and net-EndoIII sites calculated? How many cells were counted for the micronucleus assay?
RESPONSE: Thank you so much for your comments. The migration conditions and voltage gradients as well as the number of comets scored, and the program used have been included in the “Materials and methods”. The net-FPG and net-EndoIII sites were calculated subtracting the %DNA in tail value from buffer F. The number of cells counted in the distribution of bone marrow cells were 500 and for the calculation of the % of micronucleus were 5000. This information has also been clarified in the text.
- Are historical data for negative controls available for the micronucleus test? The solvent control for female rats seems high with some overlap with the positive control. How many cells were counted?
RESPONSE: Thank you for the comment. The historical data for negative controls for the micronucleus test performed in our laboratory are available in the literature: Medrano-Padial et al., 2021. Foods. The results obtained in the positive control in this study were similar to those obtained in the present work. This information about historical data of control has been added to the results: “The results obtained in the positive and negative control groups in others and recent studies [24] studies were similar to those obtained in the present work for Comet and Micronucleus assays.”
With respect to your comment about the solvent control for female rats with some overlap with the positive control, our results indicated that the control negative not showed significant differences with solvent negative. Moreover, this group induced significant differences with the positive control. The statistical results of PCE/total (female) are shown:
The information about the cells counted for the micronucleus test have been implemented in the section of Materials and methods: “The polychromatic erythrocytes (PCE) among total erythrocytes (polychromatic and normochromatic erythrocytes (NCE)) ratio and PCE for every NCE ratio were calculated by counting 500 erytrocytes per rat. The number of micronucleus found in immature erythrocytes was calculated by counting a total of 5000 micronucleated immature erythrocytes (MNPCE) per rat and were expressed as % MN”.
- The results from the comet assay are surprising. It is odd that the positive control did not generate more tail DNA considering the short time following the last exposure. As well, the percentage of tail DNA in the treated organs is very low, especially for the stomach. I am wondering if the migration conditions were not optimal. The migration time and voltage gradient needs to be specified. Are historical data for the negative controls available?
RESPONSE: Thank you for the suggestion. The information about migration conditions and voltage gradient have been specified in “Materials and methods section”. The historical data for negative controls for the comet assay performed in our laboratory are available in the recent literature: Medrano-Padial et al., 2021 (which figure in the reference list). Their results showed approximately 31% of DNA tail in positive control in stomach, similarly to around 20 % in our study. This may be due to the fact that in our laboratory we have refined the technique and we have obtained cleaner images of negative / positive controls
- The graphs for the comet assay are very difficult to interpret. An interrupted or broken y-axis is needed since the difference between the positive control and samples is so large. This would help to see any variation or trends in tail DNA.
RESPONSE: Thank you for the suggestion. In figure 2 the y-axis has been divided into two: main and secondary axis, hoping that the interpretation of the figure will be easier.
- It would also be interesting to see the variation in the distribution of comets with each treatment by presenting data for individual comets.
RESPONSE: Thank you for the suggestion. The data of comets have been implemented as supplementary material 1. In the tables you can find the data related to the comet assays, with the treatment with Buffer F, FPG and ENDOIII. A sentence has been included in the text: “The distribution of comet results with each treatment by presenting data for individual comet data is available as supplementary material.”
- It is not clear how the results for the comet assay were calculated. Were medians for each slide and animal calculated? Are the data presented the means of the medians for each group?
RESPONSE: Thank you for the suggestion. This information has been implemented in the “Materials and methods” section and the MS have been improved: “At least 150 randomly selected nuclei per rat were analyzed with the image analysis software Comet Assay IV (Perceptive Instruments, UK). %DNA in tail automatically acquired by the software, were used to define each of the comets analyzed. Endo III and Fpg sensitive sites were established by deducting the % of DNA in tail after repair enzymes incubation. The medians of the scored comets were taken to describe each animal and the means of each groups of doses and controls were represented in the Figure 2.”.
- In light of the high level of intestinal exposure, it would have been interesting to investigate intestinal effects as well, including comet and histopathological assays.
RESPONSE: Thank you for the comment. The organs taken for this study were selected taking into account the work previously carried out by other authors and experts in genotoxicity (Llana-Ruiz-Cabello et al., 2018. J. Toxicol. Env. Heal. A.; Kirkland et al., 2019. Mutat. Res.; Diez-Quijada et al., 2020. Toxins; Medrano-Padial et al., 2021. Foods). The stomach has been selected as the first organ of the digestive tract in contact with the PTS. Since there is no moderate histopathological damage and neither genetic nor oxidative damage, we do not consider that the intestine provides more information. All this information is included in the “Discussion” section: “Regarding the selection of these organs for genotoxicity assessment and histopatho-logical analysis, the liver is a primary site of xenobiotic metabolism and is often highly exposed to both parent substances and metabolites, and the stomach is a tissue of direct contact for orally administration” [30,21].
- It is interesting that the histological examination showed slight increases in damage in the liver and stomach at all concentrations tested (although only the highest concentration was statistically significant). It would have been interesting to quantify serum markers of liver damage (ALT for example)
RESPONSE: Thank you for the suggestion, it has already been taken into account. In view of the negative results obtained in these genotoxic tests for the PTS, a subchronic test (90 days) is currently being carried out, where all the biochemical (including ALT) and haematological damage markers are measured. These results will be published in the future.
- In the authors’ previous in vitro study, concentrations of PTS giving positive results in the comet assay gave rise to significant toxicity (~IC50). Could this be a false positive result based on high levels of cytotoxicity at this concentration?
RESPONSE: Thank you for your interesting question. It could be a false positive as they are very sensitive cell lines, in addition the dose tested corresponds to cytotoxic doses. As shown in the "Discussion" section: “…different sensibility of experimental model; hence, mammalian cells lines are highly sensitive to giving positive results, due to their scarcity in the metabolism, p53 function and DNA repair capability. In fact, EFSA [32] indicated that over 90% of the in vitro positives were “unrelated positives” identified by these cells under in vitro condition.”
In addition, a sentence has been included in the discussion regarding cytotoxicity as a cause of the false positive: “…This finding could be due to the fact that the in vitro tests were carried out under cytotoxic conditions
- .”Line 336: The authors indicate that 55.0 mg/kg corresponds to 331.0 μM. Does this represent serum concentration? How was this concentration calculated? Without appropriate quantification, it is extremely difficult to extrapolate. How much of the dose was absorbed? Does the concentration represent organ exposure? I would suggest to remove this extrapolation.
RESPONSE: Thank you for your suggestion. You are right, this extrapolation is difficult to comprise. The sentence has been deleted.
- Simply as a point of interest, what could be the concentration of PTS in feed or other commercial purposes necessary for antimicrobial activity compared to exposure concentrations? How does this amount correlate with the concentrations tested in this study?
RESPONSE: Thank you for your interest and for the question. The concentration to be used in the feed is 10 mg for each kilogram of feed. One farm animal of 100 kilograms eats an average of 3 kilograms of feed per day so its PTS intake would be 0.3 mg / kg b.w. This would be a dose more than 100 times lower than those tested in this study.
Overall, the present manuscript represents a substantial amount of work. The study was well-designed with an appropriate number of animals, appropriate concentrations and an appropriate treatment schedule. I agree with the authors that the in vivo results reported in the present work confirm that PTS is not genotoxic in vivo in the rat model.
RESPONSE: Thank you for your kind overall comment.
Reviewer 2 Report
Comments for the manuscript titled: Genotoxicity evaluation of Propyl-propane-thiosulfinate (PTS) by a combinate micronucleus and comet assays in rats
The authors use the combination of the in vivo micronucleus (MN) and comet assays into a single rodent study using the EFSA and OECD protocols to study the genotoxicity of propyl-propane-thiosulfinate (PTS). The authors also studied the histopathological changes in various organs upon PTS exposure. The study is relevant and the experimental design is clearly written. Their results show that PTS is not genotoxic and therefore, could be a potential alternative to synthetic preservatives and antibiotics in animal feed.
Comments:
Lines 36-38: Please clarify this statement.
Line 113: Check for spelling.
Lines 136-138: What is the rationale for the time points selected for the PTS administration? Why the sacrifice is performed 3 hours after the final dose?
Sections 2.5 ¨C 2.7: The authors should add a brief description of the experimental procedures in addition to the reference of the published protocol. This allows the reader to have a brief knowledge of the procedures and if more details are needed then he/she can look for the reference.
Line 185: Please define PCE/NCE.
Table 1: Please add the statistical test used at the footnote of the table.
Figure 2: The authors could use a graph with a combination of bars and dots (individuals) to show the distribution of DNA in the tail for each treatment group. Please add the statistical test used.
Lines 223-224: The authors state that: ¡°The histopathological study of stomach and liver did no show differences between males and females¡±. However, this analysis is not included on the manuscript. Therefore, it should be included as data not shown.
Figure 3: Based on the text, it seems that is the data from all the rats including males and females. Please state this on the figure caption. Statistical tests used should also be included. Were comparisons among treatments performed for this data? PTS doses should slightly similar scores but only the highest dose is significant when comparing to the solvent and negative control.
Figure 4: The authors should consider adding arrows to identify the areas with the histopathological changes on the images.
Additional comments: Would the effects of PTs reported by the authors vary if the exposure time is increased? What would be the long term effects of a constant exposure to PTS?

Author Response
Reviewer 2
The authors use the combination of the in vivo micronucleus (MN) and comet assays into a single rodent study using the EFSA and OECD protocols to study the genotoxicity of propyl-propane-thiosulfinate (PTS). The authors also studied the histopathological changes in various organs upon PTS exposure. The study is relevant and the experimental design is clearly written. Their results show that PTS is not genotoxic and therefore, could be a potential alternative to synthetic preservatives and antibiotics in animal feed.
RESPONSE: First of all, thank you for your thorough revision. All your suggestions have been taken into account and the MS has been improved.
Comments:
Lines 36-38: Please clarify this statement.
RESPONSE: Thank you for your comment. The sentence has been clarified.
Line 113: Check for spelling.
RESPONSE: Thank you for your remark. It has been modified in the text.
Lines 136-138:
What is the rationale for the time points selected for the PTS administration?
RESPONSE: Thank you for your question. The time points for PTS administration were selected according to the recommendations of the OECD Guideline 489. This guideline establishes that “Animals should be given daily treatments over a duration of 2 or more days (i.e. two or more treatments at approximately 24 hour intervals), and samples should be collected once at 2-6 h (or at the Tmax) after the last treatment”. Also, in agreement to the recommendations of Bowen et al., (2011), in the combination of the MN test and comet assay, animals were dosed at 0, 24 and 45 h, and samples were taken at 48 h (i.e., three hours after the final dose). For this, animal groups were dosed at 0, 24 and 45h.”
The references used for the selection criteria of the time points of administration and the scarification timing have been included in the text.
Why the sacrifice is performed 3 hours after the final dose?
RESPONSE: Thank you for your question. According to the OECD Guideline 489, sampling time in the absence of kinetic data for the measurement of genotoxicity is to sample at 2-6 h after the last treatment for two or more treatments.
And according to the criteria of Bowen et al. 2011: “These types of primary DNA lesion are short-lived and may undergo rapid repair. Therefore, in order to prevent any loss of potential DNA damage, it is recommended that sampling for DNA damage via the CMT occurs within the interval of 3–6 h after the final dose (in the case of multiple dosing)”.
For this reason, and according to our previous successful experiments in this field, we have chosen 3 hours after the final dose.
Sections 2.5 ¨C 2.7: The authors should add a brief description of the experimental procedures in addition to the reference of the published protocol. This allows the reader to have a brief knowledge of the procedures and if more details are needed then he/she can look for the reference.
RESPONSE: Thank you for the suggestion. The “Materials and methods” section has been extensively completed, comprising more information about chemical and reagents used, as well as the protocol about comet and micronucleus assay, following your suggestions as well as from the suggestion of another reviewer.
Line 185: Please define PCE/NCE.
RESPONSE: Thank you for your comment. The PCE/NCE have been defined in “Materials and methods”, section 2.5. Line 165: “The polychromatic erythrocytes (PCE) among total erythrocytes (polychromatic and normochromatic erythrocytes (NCE)) ratio and PCE for every NCE ratio were calculated by counting 500 erytrocytes per rat. The number of micronucleus found in immature erythrocytes was calculated by counting a total of 5000 micronucleated immature erythrocytes (MNPCE) per rat and were expressed as % MN”.
Table 1: Please add the statistical test used at the footnote of the table.
RESPONSE: Thank you for your suggestion. You are right. It has been included in the table.
Figure 2: The authors could use a graph with a combination of bars and dots (individuals) to show the distribution of DNA in the tail for each treatment group. Please add the statistical test used.
RESPONSE: Thank you for the suggestion. In figure 2 the y-axis has been divided into two: main and secondary axis, hoping that the interpretation of the figure will be easier. In addition, the data have been implemented in a supplementary material. The statistical test used have been implemented in figure caption.
Lines 223-224: The authors state that: °The histopathological study of stomach and liver did no show differences between males and females¡±. However, this analysis is not included on the manuscript. Therefore, it should be included as data not shown.
RESPONSE: Thanks for your suggestion. The sentence has been modified including: (data not shown).
Figure 3: Based on the text, it seems that is the data from all the rats including males and females. Please state this on the figure caption. Statistical tests used should also be included. Were comparisons among treatments performed for this data? PTS doses should slightly similar scores but only the highest dose is significant when comparing to the solvent and negative control.
RESPONSE: Thanks for your suggestions. Following your comments, the figure caption has been modified to clarify that the data are from males and females and the statistics have been included. Then, the paragraph has been modified as follows:
“Figure 3. Mean ± standard deviation (SD) of the histopathological damage recorded in liver and stomach of female and male rats treated with PTS. *Statistical difference (P<0.05) compared to the negative and solvent controls using the non-parametric Kruskal-Wallis test followed by Dunn´s multiple comparison test.”
After the revision of the data and statistics thanks to your comment, we detected a mistake in the manuscript and figure 3 regarding the statistic test applied to the histopathological data. Thus, a non-parametric Kruskal-Wallis test followed by Dunn´s multiple comparison test was carried out instead a Mann-Whitney test to compare among all groups.
As result of this checking, we realised that in the stomach the PTS 17.4 and 5.5 doses are statistically significant compared to the negative and control groups. Thus, we introduced the asterisks in the figure 3 corresponding to these treatments. Moreover, in the manuscript we introduced the word “statistically” in the line 414 to specify that these doses although had changes scored as minimal severity because the overall pathology was very slight they were statistically significant compared to the controls.
Figure 4: The authors should consider adding arrows to identify the areas with the histopathological changes on the images.
RESPONSE: Thank you for your suggestion. Symbols have been added to the histopathology figures to understand where the damages are and a legend of them has been included in the figure caption.
Additional comments: Would the effects of PTs reported by the authors vary if the exposure time is increased? What would be the long term effects of a constant exposure to PTS?
RESPONSE: Thanks for your interesting questions. In view of the negative results obtained in these genotoxic tests for the PTS, a subchronic test (90 days) is currently being carried out, where all the biochemical and haematological damage markers are being measured. Therefore, the consequences of exposure of PTS in long term will have been evidenced.

Reviewer 3 Report
The paper reports details of a high quality genotoxicity study (conducted in accordance with OECD Test Guidelines) that addresses an important question regarding the potential genotoxicity of propyl-propane-thiosulfinate. While positive responses were observed in vitro, this follow-up study indicates that PTS is not genotoxic in vivo under the conditions tested.
A few specific comments are below:
Methods
- No details are provided sections 2.5 and 2.6. the OECD test guidelines and a paper by Diez Quijada et al are referred to, but to support evaluation of the study it would be helpful to provide a few key pieces of information such as the number of erythrocytes and immature erythrocytes counted in the MN test, and the number of cells/animal that were counted in the Comet assay.
- For the comet assay, a brief description of the enzyme treatments and lysis, unwinding and electrophoresis conditions (e.g. duration of unwinding and electrophoresis) would also be helpful.
- Criteria for a positive or negative response in both assays would also be useful. Were the criteria exactly the same as those in the OECD test guidelines? For example, criteria for acceptability and a positive or negative response in the OECD test guidelines include consideration of whether the results are in line with historical negative or positive control data, but historical control data are not discussed in this paper.
Results
- Were the animals monitored for clinical signs during the study? If so it would be useful to note this in the methods and results.
- If historical control data are available, is it possible to include details in Table 1/Figure 2 or at least for them to be mentioned in the text (e.g. to indicate if they are consistent with the results in the present study)?
- Is Figure 3 a sum of all histopathological observed in the stomach and liver? This could be made clearer. This is also of questionable value compared to reporting the incidences of each specific change.
- It would be useful to include arrows or other markers in Figures 4 and 5 to indicate the histopathological changes discussed in the text.
Discussion
- Good to see bone marrow exposure addressed in the discussion.
- Measurement of clinical chemistry, haematology and urinalysis may have been helpful in interpreting the significance of the histopathological changes. Might be good to include discussion of why these weren’t included in the study or the authors’ view on if they would be useful.
- Lines 348 – 350 note that cytotoxicity can contribute to genotoxic findings in vitro. Was cytotoxicity seen in the previous in vitro studies with PTS?
- Lines 367 – 369 and lines 379 – 381: some discussion on safety for use as a feed additive and need for further studies on toxicokinetics. My understanding is that the EFSA guidance for feed additives also indicates that the basic set of toxicological studies for a feed additive should include a subchronic toxicity study (90-day). The need (or lack of) for a subchronic study should be discussed, including whether there are any plans to do such a study.
The paper is generally well written but there are a few places where the language could be more clear, plus some typos. For example:
- In the abstract (lines 15 – 16) ‘this organosulfur molecule is might used as a feed…’
- Line 99 ‘animals were humanly care in…’
- Line 70 ‘ For this purspose…’
Author Response
Reviewer 3
The paper reports details of a high-quality genotoxicity study (conducted in accordance with OECD Test Guidelines) that addresses an important question regarding the potential genotoxicity of propyl-propane-thiosulfinate. While positive responses were observed in vitro, this follow-up study indicates that PTS is not genotoxic in vivo under the conditions tested.
RESPONSE: First of all, thank you for your thorough revision. All your suggestions have been taken into account and the manuscript has been improved.
A few specific comments are below:
Methods
No details are provided sections 2.5 and 2.6. the OECD test guidelines and a paper by Diez Quijada et al are referred to, but to support evaluation of the study it would be helpful to provide a few key pieces of information such as the number of erythrocytes and immature erythrocytes counted in the MN test, and the number of cells/animal that were counted in the Comet assay.
RESPONSE: Thank you for your comment. The “Materials and methods” section has been extensively completed, comprising more information about protocols used, as well as the number of cell and nucleus counted in the micronucleus and comet assay, respectively.
For the comet assay, a brief description of the enzyme treatments and lysis, unwinding and electrophoresis conditions (e.g. duration of unwinding and electrophoresis) would also be helpful.
RESPONSE: Thank you for the suggestion. This information has been added in the “Materials and methods” including the migration conditions and voltage gradients as well as the number of comets scored, and the program used.
Criteria for a positive or negative response in both assays would also be useful. Were the criteria exactly the same as those in the OECD test guidelines? For example, criteria for acceptability and a positive or negative response in the OECD test guidelines include consideration of whether the results are in line with historical negative or positive control data, but historical control data are not discussed in this paper.
RESPONSE: Thank you for your comment. the OECD guidelines does not specify any criteria for controls. Therefore, to establish the correct values for our controls, we have relied on other authors, for example, Medrano-Padial et al. 2021 (listed in the reference list) and whose values are similar to ours. A sentence has been included in the results to clarify: The results obtained in the positive and negative control groups in others and recent studies [24] studies were similar to those obtained in the present work for Comet and Micro-nucleus assays.
Results
Were the animals monitored for clinical signs during the study? If so it would be useful to note this in the methods and results.
RESPONSE: Thank you for your suggestion. The results related to the clinical observations have been added in the “Materials and methods” and “Results” sections and the manuscript has improved. In M&M, in the section 2.3 the following sentence has been included:
“After treatment, each rat was observed twice daily for clinical signs such as changes in eye, fur or skin, changes in gait, posture and, secretions.”
Moreover, in Results section, the following sentence has been added:
“No clinical signs were observed during the complete study after treating the rats with all levels of doses of PTS, water or water added with Tween 80. However, rats treated with EMS presented piloerection and an attenuation in their normal physical activity.”
If historical control data are available, is it possible to include details in Table 1/Figure 2 or at least for them to be mentioned in the text (e.g. to indicate if they are consistent with the results in the present study)?
RESPONSE: Thank you for the comment. The historical data for negative controls for the micronucleus and comet assays performed in our laboratory are available in the literature (Medrano-Padial et al., 2021. Foods.) The results obtained in the positive control in this study were similar to those obtained in the present work. This information about historical data of control have been added to the results: “ The results obtained in the positive and negative control groups in others and recent studies [24] studies were similar to those obtained in the present work for Comet and Micronucleus assays.”
Is Figure 3 a sum of all histopathological observed in the stomach and liver? This could be made clearer. This is also of questionable value compared to reporting the incidences of each specific change.
RESPONSE: Thank you for your comments. The histopathological study was done independently in each tissue (liver and stomach separately) following a semiquantitative evaluation of the severity of the lesions observed giving the following score: 0, no significant lesions (0%); 1, minimal (<10%); 2, mild (11-25%); 3, moderate (26-50%); 4, severe (51-75%); and 5: very severe (>75%). Thus, we introduced the following phase in lines 272-274 “A semiquantitative evaluation of the severity of lesions was scored as follows: 0, no significant lesions (0%); 1, minimal (<10%); 2, mild (11-25%); 3, moderate (26-50%); 4, severe (51-75%); and 5: very severe (>75%).”
In the figure 3 the dark bar corresponds to the score obtained as a mean of the scores assigned to each animal within each group of treatment regarding the severity of the lesions of the liver in general. While the grey bar corresponds to the same treatment of the data in the stomach. Overall, the histopathology in both liver and stomach was minimal or very mild and consequently we did not consider necessary the insertion of an independent bar for the specific changes observed. Moreover, the specific changes are described in detail in the manuscript in Results and Discussion sections .
It would be useful to include arrows or other markers in Figures 4 and 5 to indicate the histopathological changes discussed in the text.
RESPONSE: Thank you for your suggestion. Symbols have been added to the histopathology figures to understand where the damages are and a legend of them has been included in the figure caption:
“Figure 4. Histopathological changes in the stomach of rats exposed to PTS. (A-C) Glandular stomach of controls (negative, solvent and positive, respectively); (D-F) Glandular stomach of PTS doses (55.0, 17.4 and 5.5 mg/kg, respectively); (G-I) Nonglandular stomach of negative control, positive control and PTS 55.0 mg/kg dose, respectively. Hematoxylin and eosin stain. Asterisk: flaking epithelium of the gastric mucosa; Nc: focal necrosis of the apical gastric glands; arrow-head: hypertrophy of the mucus cells; arrow: occasional hyperemia of the apical mucosa glands; V: vacuolation of the squamous epithelium in the mucosa; Inf: in-flammatory infiltrates in the mucosa and submucosa.
Figure 5. Histopathological changes in the liver of rats exposed to PTS. (A-C) Controls (negative, solvent and positive, respectively); (D-F) PTS doses (55.0, 17.4 and 5.5 mg/kg, respectively). Hematoxylin and eosin stain. CV: central vein; star: degenerated hepatocytes; Inf: inflammatory infiltrates.”
Discussion
Good to see bone marrow exposure addressed in the discussion.
RESPONSE: Thanks for your kind words.
Measurement of clinical chemistry, haematology and urinalysis may have been helpful in interpreting the significance of the histopathological changes. Might be good to include discussion of why these weren’t included in the study or the authors’ view on if they would be useful.
RESPONSE: Thank you for your comment. You are right, they could have been useful, but unfortunately, we did not perform such measurements since they were not required in the OECD guidelines 474, and only the OECD 489 stated that they should be useful. We appreciate your consideration, it will be taken into account in future tests, since in this all the blood samples were destined to carry out the standard and enzyme-modified comet assay.
Lines 348 – 350 note that cytotoxicity can contribute to genotoxic findings in vitro. Was cytotoxicity seen in the previous in vitro studies with PTS?
RESPONSE: Thank you for your suggestion. The idea of cytotoxicity as a contribution to genotoxic findings in vitro has been included in the text. The previous in vitro test with PTS were performed using as a highest concentration IC50 for each cell line.
Lines 367 – 369 and lines 379 – 381: some discussion on safety for use as a feed additive and need for further studies on toxicokinetics. My understanding is that the EFSA guidance for feed additives also indicates that the basic set of toxicological studies for a feed additive should include a subchronic toxicity study (90-day). The need (or lack of) for a subchronic study should be discussed, including whether there are any plans to do such a study.
RESPONSE: Thank you for your comment. The information about subchronic toxicity has been added in the Discussion: ”… In addition, following the guidance of EFSA for feed additive the subchronic toxicity should be performed through a 90-day assay.”
In view of the negative results obtained in these genotoxic tests for the PTS, a subchronic test (90 days) is currently being carried out with same dose, where all the biochemical and haematological damage markers will be measured. We hope that these results will be published in the future.
The paper is generally well written but there are a few places where the language could be more clear, plus some typos. For example:
In the abstract (lines 15 – 16) ‘this organosulfur molecule is might used as a feed…’
RESPONSE: Thank you for your comment. The sentence has been clarified.
Line 99 ‘animals were humanly care in…’
RESPONSE: Thank you for your remark. It has been modified in the text.
Line 70 ‘ For this purspose…’
RESPONSE: Thank you for your remark. It has been corrected in the text.

Round 2
Reviewer 1 Report
The authors have taken into account the comments of the three reviewers, and have adequately responded to the major concerns raised by the reviewers. Instead of the supplementary data presented in table form, I would have preferred a box plot or a scattered plot showing the distribution of comets within the group. Some minor grammatical errors have been introduced with the modifications made by the authors, however, if these small corrections in spelling can be made, the manuscript can be accepted without the need of a further round of review.
The quality of Figure 3 must be improved. A higher resolution image is necessary.
Line 45: ... and have a...
Line 47: degradates should be replaced with degraded
Line 106, 236, 334: schematic diagram
Line 115: from
Line 117: was
Line 165: scarification should be replaced with 'timing of sacrifice'
Line 187: erythrocytes
Author Response
The authors have taken into account the comments of the three reviewers, and have adequately responded to the major concerns raised by the reviewers. Instead of the supplementary data presented in table form, I would have preferred a box plot or a scattered plot showing the distribution of comets within the group. Some minor grammatical errors have been introduced with the modifications made by the authors, however, if these small corrections in spelling can be made, the manuscript can be accepted without the need of a further round of review.
The quality of Figure 3 must be improved. A higher resolution image is necessary.
Line 45: ... and have a...
Line 47: degradates should be replaced with degraded
Line 106, 236, 334: schematic diagram
Line 115: from
Line 117: was
Line 165: scarification should be replaced with 'timing of sacrifice'
Line 187: erythrocytes
RESPONSE: Thanks for your suggestions and corrections. The quality of the Figure 3 has been improved, and all the gramatical corrections have been performed.
This manuscript is a resubmission of an earlier submission. The following is a list of the peer review reports and author responses from that submission.